# The Release of Non-Extractable Ferulic Acid from Cereal By-Products by Enzyme-Assisted Hydrolysis for Possible Utilization in Green Synthesis of Silver Nanoparticles

**DOI:** 10.3390/nano12173053

**Published:** 2022-09-02

**Authors:** Vitalijs Radenkovs, Karina Juhnevica-Radenkova, Dmitrijs Jakovlevs, Peteris Zikmanis, Daiga Galina, Anda Valdovska

**Affiliations:** 1Processing and Biochemistry Department, Institute of Horticulture, LV-3701 Dobele, Latvia; 2Research Laboratory of Biotechnology, Division of Smart Technologies, Latvia University of Life Sciences and Technologies, LV-3004 Jelgava, Latvia; 3Faculty of Veterinary Medicine, Latvia University of Life Sciences and Technologies, LV-3004 Jelgava, Latvia

**Keywords:** biocomposites, biorefining, by-products, enzymatic hydrolysis, green synthesis, nano-scale objects, nanoparticles

## Abstract

The present work was undertaken to elucidate the potential contribution of biosynthetically produced ferulic acid (FA) via enzymatic hydrolysis (EH) of rye bran (RB) to the formation of silver nanoparticles (AgNPs) during green synthesis. An analytical approach accomplished by multiple reaction monitoring (MRM) using triple quadrupole mass selective detection (HPLC-ESI-TQ-MS/MS) of the obtained hydrolysate revealed a relative abundance of two isomeric forms of FA, i.e., *trans*-FA (*t*-FA) and *trans*-*iso*-FA (*t*-*iso*-FA). Further analysis utilizing high-performance liquid chromatography with refractive index (HPLC-RID) detection confirmed the effectiveness of RB EH, indicating the presence of cellulose and hemicellulose degradation products in the hydrolysate, i.e., xylose, arabinose, and glucose. The purification process by solid-phase extraction with styrene-divinylbenzene-based reversed-phase sorbent ensured up to 116.02 and 126.21 mg g^−1^ of *t*-FA and *t*-*iso*-FA in the final eluate fraction, respectively. In the green synthesis of AgNPs using synthetic *t*-FA, the formation of NPs with an average size of 56.8 nm was confirmed by scanning electron microscopy (SEM) and energy dispersive spectroscopy (EDS) techniques. The inclusion of polyvinylpyrrolidone (PVP-40) in the composition of NPs during synthesis favorably affected the morphological features, i.e., the size and shape of AgNPs, in which as big as 22.4 nm NPs were engineered. Meanwhile, nearly homogeneous round-shaped AgNPs with an average size of 16.5 nm were engineered using biosynthetically produced a mixture of *t*-FA and *t*-*iso*-FA and PVP-40 as a capping agent. The antimicrobial activity of AgNPs against Gram-positive and Gram-negative bacteria, including *Pseudomonas aeruginosa*, *E. coli*, *Enterococcus faecalis*, *Bacillus subtilis*, and *Staphylococcus aureus* was confirmed by the disk diffusion method and additionally supported by values of minimum inhibitory (MIC) and bactericidal (MBC) concentrations. Given the need to reduce problems of environmental pollution with cereal processing by-products, this study demonstrated a technological solution of RB rational use in the sustainable production of AgNPs during green synthesis. The AgNPs can be considered as active pharmaceutical ingredients (APIs) to be used for developing new antimicrobial agents and modifying therapies in treating multi-drug resistant (MDR) pathogens.

## 1. Introduction

Among the global threats this century, controlling and suppressing the spread of microbial infections has become the number one challenge. Overuse of antibiotics is the main driver leading to the development of multidrug-resistant (MDR) pathogens [1]. There is a demand for developing new antimicrobial agents and modifying therapies effective in treating MDR pathogens [2,3]. Lately, cutting-edge research in nanotechnology comes with the development of nano-scale objects with superior antimicrobial actions against MDR pathogens, indicating a platform to combat ESKAPE (*Enterococcus faecium*, *Staphylococcus aureus*, *Klebsiella pneumoniae*, *Acinetobacter baumannii*, *Pseudomonas aeruginosa*, and *Enterobacter* spp.) in the era of antimicrobial resistance (AMR) [4,5]. Among the most evolved nanotechnological applications, metallic nanoparticles (NPs) demonstrate the most unique features and versatile applications in various fields and environmental remediations [6,7,8]. In recent years, AgNPs in particular have attracted tremendous attention among the scientific community due to their superb ability to inhibit the growth of MDR bacteria and cause severe damage to cancer cells, leading to necrosis or ultimate apoptosis [9,10]. The AgNPs could be fabricated by a series of synthesis routes with varying mechanisms, inputs, yields, reactions, and resulting size and shape distributions. Recently, however, there has been an increasing preference for the green synthesis of AgNPs due to their simplicity in operation conditions and eco-friendly nature [11]. A green method for the preparation of metallic NPs involves the use of metal salts and plant-derived active compounds with reducing power [12]. Such biologically active representatives as stilbenoids [13], flavonoids [14], carotenoids [14], mono- and disaccharides [11], and essential oils [15] have been repeatedly utilized for the reduction of Ag ions NPs. Due to the availability of flavan-3-ol representatives, along with the relative stability of AgNPs, the use of green tea leaf extracts takes a leading position among other plant-based extracts used in the green synthesis of AgNPs [16,17]. The foremost reducing power of green tea extracts can be explained by the presence of bioactives containing vicinal diols of two OH groups attached to the B ring [18]. This statement is reinforced by an observation made by Radenkovs et al. [19], revealing the greatest contribution to radical scavenging activity for such compounds as proanthocyanidins, (+)-catechin, (−)-epicatechin, quercetin, and kaempferol compared with other antioxidants detected in the extracts of wild apple fruit. However, it is worth noting that the use of crude extracts in AgNPs synthesis could lead to the formation of undesirable complexes in the final products, resulting in limiting the range of AgNPs applicability [11]. This is due to the complexity of crude extracts that are replete with components of various nature. Taking this into account, more attention should be paid to the engineering of nano-scale objects using individual components, which can be obtained by either selective extraction or enzymatic hydrolysis of individual components of the sample matrix with the subsequent purification steps.

Encouraging results have been obtained by Agarwal et al. [11], demonstrating the potential utilization of individual carbohydrates such as rhamnose and agar for industrial production of AgNPs smaller than 100 nm. Meanwhile, the report of Lerma-García et al. [20] emphasizes the effectiveness of synthetic ferulic acid (FA) as the remarkable catalytic activity of gold-based NPs achieved during the reduction of 4-nitrophenol to 4-aminophenol. The attempt of Wang et al. [21] to synthesize NPs by FA as a sole reducing agent of Ag ions was unsuccessful since the formation of NPs was not observed under neutral or acidic pH conditions. Noting a “silver mirror” reaction, the authors proposed the use of NaOH, as the formation of round-shaped AgNPs-FA in an alkaline medium was achieved. Darroudi et al. [22] highlighted the need for NaOH as its plays the role of an accelerator in the formation of AgNPs in the green synthesis using 1% gelatin solution as a reducing agent. The authors reached a nearly homogeneous distribution of particle size; the average size of all prepared AgNPs was less than 20 nm, and the smaller size was 10.7 nm. However, these are the only reports demonstrating the potential utilization of this molecule in the green synthesis of AgNPs.

There is a body of clinically proven health-promoting benefits attributed to FA, including anti-inflammatory, antithrombotic, antioxidant, hepatoprotective, anticarcinogenic, metal chelation, modulation of enzyme activity, activation of transcriptional factors, etc. [23]. Furthermore, superior activity to combat MDR bacteria has been repeatedly demonstrated [24,25]. In an in vivo study, the incorporation of FA together with the other hydroxycinnamic acid derivatives in sunscreens improved the photoprotective functions of the product by 37% and reduced skin inflammatory reactions [26]. The aforementioned positive features of FA make this molecule a good candidate to be used as a reducing agent in the green synthesis of AgNPs. It is worth noting, though, that a lack of sustainable and cost-effective alternatives to FA production and challenges that currently persist during its extraction from natural sources affect the price of this compound.

FA or 3-(4-hydroxy-3-methoxyphenyl)acrylic acid is the representative of hydroxycinnamic acid produced largely by chemical synthesis routes at an industrial scale. It has been found that in cereal by-products, FA in 90–95% is covalently bound to other macromolecules such as arabinoxylans, which make this molecule non-extractable from the plant matrix while employing conventional solvent extraction using methanol, ethanol, or acetone [27]. Generally, the transition of FA from the matrix to the solvent does not occur or occurs inefficiently. Isolation attempts with conventional liquid-liquid or solid-liquid extraction (ultrasonic treatment, maceration, Soxhlet-type extraction, hydrodistillation) do not deliver desired FA yields. The effectiveness of alkaline hydrolysis in the partial or complete hydrolysis of cellulose and hemicellulose with simultaneous formation/ release of small molecular compounds (mono- and oligosaccharides, amino acids, phenolic acids, etc.) from the cereal bran (CB) matrix has repeatedly been confirmed [28,29]. Considering the issue of environmental pollution, though, the use of alkalis in the pre-treatment is not environmentally feasible.

Very recently, a more sustainable method of FA release was developed by Radenkovs et al. [30] that allows up to 369.3 and 255.1 mg of FA release from 100 g^−1^ rye (RB) and wheat bran (WB) after 48 h of EH, respectively. The ability of cellulose-degrading enzymes (C-DE) to act synergistically on the dietary fiber of CB and to catalyze the cleavage of ester bonds between arabinoxylans and bioactives (FA specifically) in WB and RB was identified. This observation, along with intrinsic biological functions of FA that perhaps would be inherited by AgNPs itself, promoted the design of this study focusing on the isolation of FA from far too problematic agro-industrial side stream viz bran and demonstration of its further use as a reducing agent in green AgNPs synthesis.

## 2. Materials and Methods

### 2.1. Plant Material

Commercial food-grade rye bran (RB) (*Secale cereále* L.) was obtained from a local supplier, “Voldemars Ltd.”. According to morphological assessment, hydrological layers such as the inner pericarp (tube cells, cross cells), outer pericarp, aleurone, and hyaline layers with attached starch granules and seed coat (testa) were distinguished in the bran sample. The approximate composition of the RB sample is shown in Table 1.

### 2.2. Plant Material Preparation for Enzyme-Assisted Hydrolysis

RB sample before EH was ground to reach Ø 0.5 mm particle size using the water-cooled “KN 295 Knifetec™” rotor mill (FOSS, Hilleroed, Denmark). Inactivation of native microorganisms and enzymes was conducted by mixing RB sample with ultrapure water (UPW) at a ratio of 1:10 *w*/*v* in 50.0 mL reagent bottles with screw caps (VWR™ International, GmbH, Darmstadt, Germany), followed by autoclaving using a “Raypa, AES 110” (Barcelona, Spain) digital autoclave with counter-pressure for 10 min at 121 ± 1 °C and a counter-pressure of 2.0 Pa. After thermal conditioning, the liquid fraction was decanted, while solids were freeze-dried using a “HyperCOOL, HC3110” freeze-drying system (Hanil Scientific Inc., Gimpo, Korea) at −110 ± 1 °C under a vacuum of 0.056–0.070 mBar for 48 h. Dried solids were packed in polypropylene zip-lock silver bags (high-density polyethylene polymer, density 3 mm, Impak Co., Los Angeles, CA, USA) (200 g in each) and stored at −18 ± 1 °C temperature until further analysis, and used for a maximum of two weeks. Moisture content was analyzed gravimetrically, as proposed by Ruiz [31].

### 2.3. Chemicals and Reagents

Commercial standards, i.e., *trans*-isomer of ferulic acid (*t*-FA) and *trans*-isomer of *iso*-FA (*t*-*iso*-FA), xylose (Xyl), arabinose (Ara), fructose (Fru), glucose (Glu), sucrose (Suc), maltose (Mal), and glycerol (Gly), were purchased from Sigma-Aldrich Chemie Ltd., (Steinheim, Germany). Sodium hydroxide (NaOH), citric acid (C_6_H_8_O_7_), 2,2-diphenyl-1-picrylhydrazyl, sodium citrate dihydrate (C_6_H_5_Na_3_O_7_·2H_2_O), polyvinylpyrrolidone (PVP-40), of liquid chromatography-mass spectrometry (LC-MS) grade ethanol (EtOH), methanol (MeOH), and formic acid (HCOOH) (puriss p.a., ≥99.9%) were purchased from Merck KGaA (Darmstadt, Germany). Silver nitrate (AgNO_3_) (puriss p.a., ≥99.9%) was obtained from Chempur (Piekary Śląskie, Silesia, Poland). Mueller-Hinton broth was purchased from Oxoid, Inc., Thermo Scientific™ (Hampshire, United Kingdom), while Mueller-Hinton Agar II was acquired from Biolife Italiana, Mascia Brunelli (Milano, Italy). The UPW was produced using the reverse osmosis PureLab Flex Elga water purification system (Veolia Water Technologies, Paris, France).

### 2.4. Enzymes

Commercially available food-grade C-DE preparation Viscozyme L containing a wide range of carbohydrases was provided in kind by the company “Novozymes^®^” (Bagsvaerd, Denmark) for laboratory purposes (Table 2).

### 2.5. Preparation of the Hydrolysates by Enzyme-Assisted Hydrolysis of Rye Bran

The E-AH of RB sample utilizing commercial multi-enzyme preparation was carried out in a shaking water bath “SW23” from Julabo^®^ with a capacity of 20.0 L (Saalbach-Hinterglemm, Germany). The conditions for E-AH were selected following the recommendations of Novozymes^®^ and supporting the protocol described by Juhnevica-Radenkova et al. [32]. The E-AH of non-starch polysaccharides, i.e., hemicellulose and cellulose, was conducted using multi-enzyme complex Viscozyme L. To this purpose, 30.0 mL 0.5 M sodium citrate buffer with a pH of 4.6 containing 1.6 FBG mL^−1^ of endo-1,3-(1,4)-β-D-glucanase was added to 3.0 g RB sample. The mixture was then vortexed for 2 min using the “ZX3” vortex mixer (Velp^®^ Scientifica, Usmate Velate, Italy) and incubated for 48 h in a water bath at 44 ± 1 °C and 100 rpm. After EH, the reaction was terminated by subjecting obtained hydrolysates to thermal processing for 10 min at 99 ± 1 °C.

### 2.6. Purification of trans-FA and trans-iso-FA from Rye Bran Hydrolysates Using Solid-Phase Extraction with Strata-X and Supel^TM^-Select HLB Columns

#### 2.6.1. Small-Scale Purification

In a small-scale solid-phase extraction (SPE) process, isolation and purification of *t*-FA and *t*-*iso*-FA from RB hydrolysates were performed following protocols provided by Phenomenex and Supelco with minor modifications. Briefly, 3.0 mL of bran hydrolysate was centrifuged at 10,000 rpm (10,280× *g*) for 10 min at 20 ± 1 °C in a “Hermle Z 36 HK” centrifuge (Hermle Labortechnik, GmbH, Wehingen, Germany). The obtained supernatant was decanted and filtrated through a 0.45 µm polyvinylidene fluoride (PVDF) hydrophilic membrane filter (Durapore, Millipore, Billerica, MA, USA). Purification of *t*-FA and *t*-*iso*-FA was carried out using SPE “Strata-X” column (Phenomenex, Torrance, CA, USA) packed with a surface-modified styrene skeleton with a pyrrolidone group sorbent (33 µm, 85 Å, 30 mg 3.0 mL^−1^). An alternative SPE purification method utilizing “Supel^TM^-Select HLB” (Supelco, Bellefonte, PA, USA) column packed with a hydrophilic modified, styrene-based sorbent (50–70 μm, 80–200 Å, 60 mg 3.0 mL) was applied to comparative purpose. A constant flow (1.0 ± 0.2 mL min^−1^) during analytes desorption was ensured by a “Chromabond^®^ SPE” (Düren, Germany) SPE vacuum manifold with an adjusted pressure of –1.0 ± 0.1 ″Hg.

The conditioning/equilibration of SPE columns was conducted using 1 column volume (3.0 mL) of pure MeOH, followed by 1 column volume of UPW. The loaded hydrolysate (3.0 mL) was washed with 2 column volumes of UPW. In both cases, the flow-through fractions were collected for qualitative and quantitative chromatographic analysis of the presence of *t*-FA and *t*-*iso*-FA and mono- and disaccharides. The 1.0 mL acidified 50% EtOH solution (EtOH:H_2_O:HCOOH at ratio of 50:48:2 *v*/*v*/*v*) was used as an eluate for desorbing *t*-FA and *t*-*iso*-FA from polymers. The resulting fractions were collected and analyzed with an HPLC-ESI-TQ-MS/MS HPLC system.

#### 2.6.2. Large-Scale Purification

In a large-scale SPE process, a “Strata-X” column with a larger volume and mass of the packed sorbent (33 µm, 85 Å, 500 mg 12.0 mL^−1^) was used. For *t*-FA and *t*-*iso*-FA purification, 100.0 mL of bran hydrolysate was prepared similarly to a small-scale purification. Meanwhile, conditioning/equilibration of the column was conducted using 12.0 mL pure MeOH, followed by 12 mL UPW. The loaded sample was washed with 2 column volumes of UPW (24.0 mL), and a flow-through fraction was collected for further chromatographic analysis and work. Analyte elution was conducted using 1 volume (12.0 mL) acidified 50% EtOH (EtOH:H_2_O:formic acid ratio 50:48:2 *v*/*v*/*v*) at a flow rate of 3.0 mL min^−1^ under a pressure of –5.0 ± 0.1 Hg. The resulting EtOH eluate fraction was freeze-dried at −110 ± 1 °C under a vacuum of 0.056–0.070 mBar for 48 h. Further isolation of *t*-FA and *t*-*iso*-FA (present as sodium ferulate) was accomplished by solid-liquid extraction using MeOH as a sole extractant. The obtained dry residues (~5.0 g) were dissolved in 12.0 mL MeOH and subjected to ultrasonication at 50 kHz with an output wattage of 360 W for 10 min at 25 ± 1 °C using an Ultrasons ultrasonic bath (J.P. Selecta^®^, Barcelona, Spain). The mixture was double centrifuged at 10,280× *g* for 20 min at 25 ± 1 °C and supernatants were collected and filtrated through a 0.20 µm hydrophilized polytetrafluoroethylene (H-PTFE) membrane filter (Macherey-Nagel GmbH & Co. KG, Dueren, Germany). A “Laborota 4002” rotary evaporator (Heidolph, Swabia, Germany) operating at 250.0 ± 2.0 mBar and 50 ± 1 °C was utilized for complete removal of MeOH fraction. The resulting dry residue was stored at a temperature of −18 ± 1 °C until further analysis and used for a maximum of one week. A schematic representation of sample preparation steps is depicted in Figure 1.

### 2.7. Preparation of Silver Nanoparticles Using Rye Bran-Derived Mixture of trans-FA and trans-iso-FA

The stock solution with the molarity of 100 mM AgNO_3_ was used as the source of AgNPs. For the preparation of control AgNPs using synthetic *t*-FA, 9.850 mL of UPW containing 10.0 mg *t*-FA was heated in 50.0 mL reagent bottles with a screw cap for 5 min at 80 ± 1 °C and continuous stirring at 450 rpm. Subsequently, the 50.0 µL 1.0 M NaOH was added through the upper opening, and the solution was allowed to mix well. Afterwards, 0.1 mL of 100 mM AgNO_3_ solution was added drop-wise, thus ensuring 1 mM AgNO_3_ in the final solution. In this study, the NaOH was used as an accelerator in the green synthesis of AgNPs as proposed by Darroudi et al. [22]. The same protocol was used for the synthesis of AgNPs using RB-derived biosynthetic *t*-FA and *t*-*iso*-FA, where the amount of synthetic *t*-FA was substituted with 20 mg of FA-rich fraction containing 2.3 and 2.5 mg of *t*-FA and *t*-*iso*-FA, respectively.

In a separate trial, the effect of the capping agent was clarified by the addition of 80 mg amphiphilic polyvinylpyrrolidone (PVP-40) agent to the mixture of AgNO_3_ and reducing compound while keeping the same proportions.

The reaction mixtures were exposed to heating for 1 h at 80 ± 1 °C, ensuring continuous stirring at 450 rpm using a magnetic stirrer. The mixture gradually turned from light yellow to reddish-brown color during thermal exposure over 1 h. The whole reaction was carried out in the dark, covering the vessel with foil to preserve the AgNO_3_ from aggregation. The obtained suspensions of AgNPs-FA were centrifuged at 10,000 rpm (10,280× *g*) for 50 min at 20 ± 1 °C. The pellet containing AgNPs was washed 3–4 times with UPW to remove free silver ions. The precipitated AgNPs were freeze-dried at −110 ± 1 °C under a vacuum of 0.056–0.070 mBar for 24 h. Lyophilized NPs were stored in a cool, dry, and dark place and further used for their characterization. A schematic representation of AgNPs preparation steps following the green synthesis principle is shown in Figure 2.

### 2.8. Preparation of Standard Stock Solution

Stock solutions containing either 10.70 mg of *t*-FA or 2.77 mg of *t*-*iso*-FA were prepared separately in 10.0 mL 100% acidified MeOH solution (MeOH:HCOOH ratio of 99:1 *v*/*v*). Quantification of compounds was conducted by injecting 3.0 μL at 10 °C of respective calibration solution within the range of 0.0107 to 0.1177 and 0.00831 to 0.277 mg mL^−1^ of *t*-FA and *t*-*iso*-FA, respectively. Regression coefficients (R^2^) and values for the limit of detection (LOD) and quantification (LOQ) are given in Table 3. The working solution was prepared immediately before being used.

### 2.9. The HPLC-ESI-TQ-MS/MS Analytical Conditions for Ferulic Acid

The analyses were carried out using a Shimadzu series Nexera UC SFC-SFE-LC system (Tokyo, Japan) coupled to TQ-MS-8050 (Tokyo, Japan) with an electrospray ionization interface (ESI). A sample of 3.0 μL was injected onto a reversed-phase Shim-Pack UC-RP column (5.0 μm, 250 × 4.6 mm; Tokyo, Japan) operating at 45 °C and a flow rate of 1.0 mL min^−1^. Separation of compounds was conducted using an isocratic mobile phase composed of 80% acidified MeOH (1% formic acid, *v*/*v*) (A) and 20% acidified UPW (1% formic acid, *v*/*v*) (B) (*v*/*v*). The MeOH injections were included in every two samples as a blank run to avoid the carry-over effect. Data were acquired using LabSolutions Insight software, which was also used for instrument control and processing. The ionization in both positive and negative ion polarity modes was applied in this study, while data were collected in profile and centroid modes, with a data storage threshold of 5000 absorbance for MS. The operating conditions were as follows: detector voltage 1.8 kV, conversion dynode voltage 10.0 kV, interface voltage 4.0 kV, interface temperature 300 °C, desolvation line temperature 250 °C, heat block temperature 400 °C, nebulizing gas argon (Ar, purity 99.9%,) at flow 3.0 L min^−1^, heating gas carbon dioxide (CO_2_, purity 99.0%,) at flow 10.0 L min^−1^, and drying gas nitrogen (N_2_, separated from air using a nitrogen generator system from Peak Scientific Instruments Ltd. (Inchinnan, Scotland, UK), purity 99.0%) at flow 10.0 L min^−1^. Compounds were observed in the programmed and optimized multiple reaction monitoring (MRM) mode. The MRM transitions, collision energy, Q1, Q3, and dwell time for *t*-FA and *t*-*iso*-FA are depicted in Table 3 and Appendix A.

### 2.10. The HPLC-RID Conditions for Carbohydrates Analysis

Quantitative analysis of mono- and disaccharides in hydrolysate after EH and wash-through fraction after SPE was conducted using a “Waters Alliance” HPLC system (model No. e2695) coupled to a 2414 RI detector and a 2998 column heater (Waters Corporation, Milford, MA, USA) following the methodology described by Radenkovs et al. [30].

### 2.11. Scanning Electron Microscope (SEM) and Energy Dispersive Spectroscopy (EDX)

The samples of AgNPs were observed using a Mira3 scanning electron microscope (SEM) by Tescan Orsay Holding, a.s. (Brno-Kohoutovice, Czech Republic). Freeze-dried AgNPs were mounted in a thin layer onto SEM pin stubs using double-sided adhesive carbon discs, and non-fixed AgNPs were blown by a gentle stream of N_2_. The conditions were set up to operate under high vacuum mode using the Back-Scattered Electron (BSE) and Secondary Electron (SE) detectors mixer. The AgNPs were analyzed by increasing magnification up to 250 kx for precise dimension measurement and elemental composition analysis operating at 15 kV acceleration voltage. Elemental composition analysis of the AgNPs was carried out with INCA x-act LN2-free Analytical Silicon Drift Detector with PentaFET^®^ Precision energy-dispersive X-ray spectrometer by Oxford Instruments Inc. (Bognor Regis, UK), operating at 15 mm working distance and dead time in the range from 40% to 60% for energy spectrum accumulation.

### 2.12. In Vitro Susceptibility Tests

#### 2.12.1. Minimum Inhibitory Concentration (MIC)

The minimum inhibitory concentration (MIC) was estimated using the microdilution method in 96-well plates [30]. Five standard strains of Gram-negative and Gram-positive test cultures were used for antimicrobial sensitivity testing, including *Pseudomonas aeruginosa* ATCC10145, *Escherichia coli* ATCC25922, *Enterococcus faecalis* ATCC29212, *Bacillus subtilis* ATCC6633, and *Staphylococcus aureus* ATCC 6538P. Bacteria were grown on Nutrient agar (NA, Oxoid, Inc., ThermoFisher Scientific, Hampshire, UK) for 20 ± 2 h at 36 ± 1 °C and then re-suspended in Mueller-Hinton broth (MHB, Oxoid, Inc., Thermo Scientific™, Hampshire, UK). The suspension was adjusted to the final turbidity of 0.5 McFarland units (10^8^ CFU mL^−1^) using DEN-1B McFarland Tube Densitometer (Grant Instruments Ltd., Cambridge, UK). The suspension was subsequently diluted to a concentration of pprox. 1.5 × 10^6^ CFU mL^−1^ (bacterial inoculum). Under aseptic conditions, the tested AgNPs, along with biosynthetic RB-derived *t*-FA and *t*-*iso*-FA and commercial *t*-FA, were dissolved in 1 mL UPW separately over the range from 11.0 to 100.0 mg mL^−1^ (stock solution). To identify the concentration-dependent antimicrobial activity of engineered AgNPs, a serial two-fold dilution technique was used. Briefly, the concentration of the active compound in the first well was adjusted to 25% by mixing 100.0 µL active compound (from stock solution) with 100.0 µL MHB and 100 µL of prepared bacterial inoculum. The negative control consisted of MHB, while the positive control of MHB and bacterial inoculum consisted of test cultures. To calculate MBC, before incubation of microplates, 1 µL of each positive control aliquot was inoculated into two Mueller-Hinton Agar II (MHA, Biolife, Milano, Italy) plates for baseline concentration of the tested bacteria. After incubation for 24 h at 36 ± 1 °C, the MIC was defined as the lowest concentration resulting in inhibiting bacterial growth based on the absence of visible turbidity [33].

#### 2.12.2. Minimum Bactericidal Concentration (MBC)

The MBC was used as complementary to MIC assay to identify the lowest concentration of engineered AgNPs leading to bacteria death. For this purpose, an aliquot of 1 µL from all the wells showing no visible bacterial growth was subcultured into MHA and incubated for 24 h at 36 ± 1 °C. Afterward, the colony-forming units were enumerated and the MBC was defined as the lowest concentration of AgNPs leading to 95% and 99.5% bacteria death.

#### 2.12.3. Disk Diffusion Method

The antibacterial activity of AgNPs against selected pathogens was carried out using the Kirby–Bauer disk diffusion susceptibility test. Briefly, the bacteria strains (in concentration 0.5 McFarland) were spread on the (MHA) using a sterile cotton swab. The sterile paper disks (with a diameter 6 mm) were placed on the agar plate and soaked with 10 μL of biosynthetically obtained mixture of *t*-FA and *t*-*iso*-FA (brown fraction FA-RB) along with commercially available *t*-FA (FA-ST) both dissolved in 1 mL UPW, the solution containing 11.0 mg mL^−1^ of AgNPs with the average size of 16.5 nm engineered by biosynthetic *t*-FA and *t*-*iso*-FA (FA-RB-AgNPs) and AgNPs with the average size of 22.4 nm engineered by synthetic *t*-FA (FA-ST-AgNPs). The plates were incubated for 24 h at 36 ± 1 °C, and the zone of inhibition was observed and measured in mm.

### 2.13. Statistical Analysis

The results obtained are shown as means ± standard deviation of three replicates (*n* = 3). A *p*-value of ≤ 0.05 was used to denote significant differences between mean values determined using one-way analysis of variance (ANOVA) and Duncan’s multiple range test performed using IBM^®^ SPSS^®^ Statistics version 20.0 (SPSS Inc., Chicago, IL, USA).

## 3. Results and Discussion

### 3.1. Release of trans-Ferulic Acid trans-iso-Ferulic Acids from Rye Bran Utilizing Enzyme-Assisted Hydrolysis

Hydroxycinnamic acid derivatives comprise a large group of simple phenolic acids found mainly in cereals, fruit, and vegetables [34]. Due to potent antioxidant and anti-inflammatory activities and bioavailability, these compounds received tremendous attention in the field of biochemistry and pharmacology [35]. The report of Pazo-Cepeda et al. [36] reveals that FA is the most abundant representative of phenolic acids found in WB. It is worth noting, though, that nearly 97% of FA in CB is represented in bound form with arabinoxylans and other macromolecules, making the correct estimation of this molecule challenging. In the quantitative analysis of phenolic compounds using HPLC-ESI-TQ-MS/MS, Radenkovs et al. [30] highlighted the dominance of FA, making a 90% contribution to the total phenolics found in WB and RB hydrolysates. EH was found to be an environmentally friendly alternative to alkaline hydrolysis that can represent the future for sustainable industrial-scale FA production.

Supporting the “Green Deal” concept, in this experiment, a brown liquid (crude hydrolysate) with a distinct aroma of cloves and baked bread was produced as a result of RB EH with Viscozyme L for 48 h. Selective HPLC-ESI-TQ-MRM-MS/MS analysis confirmed the presence of *t*-FA and *t*-*iso*-FA with MRM transitions of 192.9500→134.0000 and 194.9000→177.1500, respectively. The amount of *t*-FA and *t*-*iso*-FA was found to be 6.48 and 14.50 mg 100 mL^−1^ (Figure 3A). Further small-scale SPE process utilizing either Strata X or Supel™-Select HLB columns ensured up to a 1.86 and 1.74–fold increase in the concentration of *t*-FA, and *t*-*iso*-FA in the eluate fraction, by applying 1.0 mL acidified 50% EtOH as an eluent. Even though both columns are packed with surface-modified styrene sorbents, a statistically higher recovery of *t*-FA and *t*-*iso*-FA was achieved using the Strata-X column having a lower mass of the polymer. The presence of pyrrolidone group in Strata-X column coupled with modified styrene skeleton ensured multiple modes of retention, including hydrophobic and π- π interactions as well as hydrogen bonding. A fairly similar observation was made by Čizmić et al. [37], highlighting the advantage of Strata-X over other SPE columns used.

Further large-scale SPE process utilizing exclusively Strata-X column with larger volume and filled mass of the polymer delivered satisfactory results. Under a large sample load, 500 mg Strata-X ensured statistically better results, as a 2.13-fold higher average concentration of FA has been observed in the eluate fraction after 12.0 mL hydrolysate load (Figure 3B). This is reinforced by an observation made by Čizmić et al. [37] and also by Mcdowall and Massart [38] revealing that due to a larger sorbent mass, the number of active sites increases, resulting in a higher analyte yield percentage. Further loading of the sorbent, considering 10–15% *w*/*w* of the sorbent mass as the maximum load by the analyte, 100.0 mL of hydrolysate was applied (20.98 mg sum of *t*-FA and *t*-*iso*-FA) to Strata-X. Up to 3.27-fold higher average FA yield was obtained compared with the initial concentration and 1.54-fold higher than with 12.0 mL hydrolysate load.

The recovery of *t*-FA from 30 mg Strata X and 60 mg Supel™-Select HLB sorbents reached 74.4% and 72.4% compared to the initial amount of FA, respectively (Figure 3C). Meanwhile, the recovery of *t*-*iso*-FA was 11.9% and 11.0% higher than that of *t*-FA. A higher value of *t*-*iso*-FA is due to its better solubility in EtOH rather than in H_2_O. This statement was reinforced by Sun et al. [39], observing a relatively better extractability of *t*-*iso*-FA with 95% EtOH than with 50% EtOH. Further analysis revealed that larger sorbent mass and sample loading volume could ensure comparable or even better recovery of FA than during small-scale SPE. As seen, the recovery of *t*-FA from the 500 mg Strata-X column with 12.0 mL volume of hydrolysate loading was 4.2% higher compared with the 30 mg Strata-X column, while the recovery of *t*-*iso*-FA was 1.1% lower. Loading the sorbent with 100.0 mL hydrolysate, the yield of *t*-FA was 6.0% higher in comparison with the yield obtained during small-scale SPE using 30 mg Strata-X. However, no statistically significant difference was found between the yield of *t*-*iso*-FA obtained during either small or large-scale SPE. Quantitative analysis of mono- and disaccharides using an HPLC-RID system revealed no presence of any sugars in the eluate fraction obtained either with Strata X” or Supel™-Select HLB columns, indicating that the method is suitable for complete removal of sugars negatively affecting ionization efficiency along with reducing the signal intensity of monomeric phenolic compounds. A similar observation has been given by De Villiers et al. [40], indicating the effectiveness of styrene-divinylbenzene-based SPE cartridges in the analysis of low molecular weight polyphenols.

### 3.2. Release of Mono- and Disaccharides during Enzyme-Assisted Hydrolysis of Rye Bran

The previous experiment revealed the effectiveness of selected hydrolytic multi-enzyme complex to release bound FA from RB within a single 48 h run of EH. Therefore, further work aimed to assess the amount of mono- and disaccharides in hydrolysate and flow-through fraction collected after 48 h of EH and SPE, respectively. A detailed profile of individual sugars after EH is depicted in Table 4 and the Appendix A. Glucose, xylose, arabinose, and fructose were the major products released from RB after EH.

Glucose is the main degradation product of cellulose and was found to be the prevalent aldohexoses representative in RB hydrolysate. This observation is in line with Shang et al. [41], though the concentration found in this study was many folds higher. Such a difference could be a case of either an analytical approach used for quantitative analysis of glucose or selectivity and catalytic activity of enzymes applied for EH. As seen, up to 84.11% of the total glucose was observed in the flow-through fraction obtained with the Strata-X SPE column. Moreover, the further study did not reveal the presence of glucose in the eluate fraction, indicating that the residual glucose was effectively desorbed from sorbent during a wash-up step.

Xylose was the main aldopentose detected in RB hydrolysate after EH, and the concentration of this monomer was 16.3-fold higher compared with the untreated RB. The rise in xylose in RB hydrolysate indicates the xylanolytic activity of multi-enzyme preparation used in this study and the capability of degrading xylosidic bonds within the xylan backbone chain in arabinoxylans, causing a simultaneous rise in xylose. This statement could be reinforced by an observation made by Ping et al. [42] pointing out an effective conversion of corn strove hemicelluloses into xylose. The SPE made it possible to purify compounds of interest since 93.87% of the total xylose was observed in the flow-through fraction after SPE.

A relatively lower, but still relevant amount of arabinose was released after EH of RB. The concentration of this aldopentose was found to be 4-fold higher compared with the untreated RB sample. A similar observation was made by Escarnot et al. [43] and just recently by Radenkovs et al. [30], indicating direct involvement of α-L-arabinofuranosidase resulting in the splitting of terminal α-L-arabinofuranoside residues in α-L-arabinosides with the simultaneous formation smaller xylan polysaccharides of feruloylated arabinose monomers.

No free fructose was detected in untreated RB since this representative in CB is found mainly as a linear or branched fructan polymer composed of fructose monomers linked with either β-(1→2) or β-(2→6) bonds and containing one or no internal or terminal glucose unit. The concentration of fructan in RB is about half higher than in WB, corresponding to 6.0–7.2% [44,45]. Park et al. [46] demonstrated the efficient release of fructose and glucose from rice straw with the use of two-step pretreatment accomplished by thermal processing and EH with a mixture of cellulase and saccharification enzymes. This observation made it attainable to assume that the multi-enzyme complex used in this study contributed to partial hydrolysis of either fructan polymers or oligomers with the simultaneous release of fructose monomers. Meanwhile, mild acidic conditions ensured by sodium citrate buffer promoted hydrolysis of acid-hydrolyzable sugars such as sucrose into glucose and fructose. Interestingly, the presence of glycerol, lactose, and galactose was also confirmed in bran hydrolysates. However, part of the supplementary research with direct enzyme injections revealed the abundance of these sugars as part of commercial enzyme preparation Viscozyme L.

To ensure the green synthesis of AgNPs, the eluate fractions collected after SPE were combined and subjected to freeze-drying for 48 h for complete removal of H_2_O and EtOH. This approach allowed the obtainment of dry residue with the concentration of *t*-FA and *t*-*iso*-FA corresponding to 143.69 and 145.15 mg g^−1^, respectively (Figure 4).

The obtained fraction was poorly soluble in MeOH, which allowed us to speculate the possible presence of sodium salt which has remained in the eluate fraction after SPE. An isolation attempt aimed at extracting compounds of interest with solid-liquid extraction using MeOH as a sole solvent, ensured 116.02 and 126.21 mg g^−1^ of *t*-FA and *t*-*iso*-FA in the final syrup-like fraction, respectively.

### 3.3. Changes in Morphology of the Rye Bran Affected by Enzyme-Assisted Hydrolysis

The assessment of morphological changes in native and EH RB samples was conducted by SEM analysis (Figure 5). As seen, a nearly homogeneous epidermal layer with no obvious cracking or decomposition was observed in the untreated RB sample (Figure 5B,C). The presence of two starch fractions was distinguished in starchy endosperm (Figure 5D). Starch granules of spherical shape with a size smaller than 10 µm take prevalence over ellipse-shaped granules with sizes of 30–50 µm, making a strong carcass those integrated into the matrix of starchy endosperm. Further analysis revealed partial fracture and decomposition of RB epidermal layer after EH for 48 h with multi-enzyme complex Viscozyme L (Figure 5C1). Meanwhile, visible ellipse-shaped void spaces of approximately 40–50 µm in diameter were observed within starchy endosperm matrix (Figure 5D1). A similar observation in disruption of matrix integrity induced by hydrolytic enzymes was revealed for wood fibers [47] and for wheat straw [48], for more efficient hydrolysis; however, the latter authors proposed using 4% sulfuric acid as pre-treatment necessary for delignification. The observed morphological changes of RB are due to the ability of cellulolytic and xylanolytic enzymes present in multi-enzyme complex Viscozyme L to act specifically on endo-1,3-(1,4)-β-D-glycosidic bonds in cellulose and on 1,4-β-D-xylosidic bonds in arabinose-containing hemicelluloses, releasing relatively shorter polymers, oligomers, and monomers. This statement could be reinforced by [49].

### 3.4. Structural Analysis of Silver Nanoparticles

It has been reported that the presence of OH and COOH groups in the structure of FA makes it possible to use it in reaction with AgNO_3_ both as a reducing agent and as a stabilizer. In the redox reaction, AgNPs with an average size of 28–33 nm could be engineered along with the formation of yellow coloration of the solution, indicating the presence of FA degradation products: quinones [50]. In this study, however, neither coloration nor formation of NPs was observed in the course of the green synthesis of AgNPs with pure synthetic *t*-FA under neutral conditions. Apparently, due to lacking oxygenated functionalities, the ability to chelate metal ions by pure *t*-FA was less pronounced under neutral conditions. However, a further experiment under alkaline conditions by adding 50.0 µL 1.0 M NaOH to the working solution ensured a gradual color change from light yellow to reddish-brown during thermal exposure for 1 h, revealing the formation AgNPs. A similar observation was made by Wang et al. [21], indicating a “silver mirror” reaction and lack of NPs. Meanwhile, considerable intensification of AgNPs plasmon resonance light scattering was observed after alkalization of working solution with NaOH. Encouraging results were achieved by Darroudi et al. [22], highlighting the role of NaOH as an accelerator favorably influencing the speed of green synthesis and morphological features of obtained AgNPs. A plausible explanation for achieving an enhanced synthesis of AgNPs under alkaline was provided by Melkamu and Bitew [6], pointing out the changes in the dissociation constant values of the functional groups present in the biomolecules involved in the reduction of Ag^+^ ions as a function of pH increase. The analysis of green synthesized AgNPs microstructure allowed the formation of NPs to be confirmed, as shown in Figure 6A. The average size of NPs with irregular shape and size distribution was found to be 56.8 ± 15.0 nm. The presence of AgNPs was confirmed by EDS analysis, showing distinctive signal peaks within the range of 3.0–3.4 keV due to surface plasmon resonance [51]. Meanwhile, the EDX pattern at two randomly selected regions revealed the presence of sulfur (S) and copper (Cu) elements, the contribution of which was found to be below 1% of the weight, indicating relative purity of obtained AgNPs (Figure 6D1). The presence of C (carbon) as the second prevalent element observed along with Ag is conditioned by the double-sided adhesive carbon disc used for the fixation of AgNPs. It has been proposed that amphiphilic PVP can act as stabilizing and/or reducing agents alone or in combination with specific functional molecules and promote the formation of NPs with a narrow particle size distribution of an average of 17 nm [52]. This statement has also been reinforced by this study’s results, by observing the homogeneous shapes and substantially smaller size of PVP capped AgNPs corresponding to an average size of 22.4 ± 7.0 nm. However, in some cases, the minor aggregation has also been identified, perhaps due to the “depletion flocculation” phenomenon caused by a high concentration of PVP in the solution [53].

In the next study, the inclusion of RB-derived fraction rich in *t*-FA and *t*-*iso*-FA in the working solution was tested as a reducing agent in the green synthesis of AgNPs. The results implied a substantial contribution of these two hydroxycinnamic acid representatives to the formation of round-shaped AgNPs with narrow size distribution, corresponding to 16.5 ± 3.2 nm on average. The reduction in the size of AgNPs is perhaps due to additional hydroxyl moieties ensured by the presence of *t*-*iso*-FA, which also was directly involved in reducing Ag ions to NPs. This observation is in line with that of Timakwe et al. [54], reporting a substantial reduction of the size of AgNPs from 65 to 37.5 nm achieved by supplementation of working solution with the additional amount or reducing agent.

As the size and shape of AgNPs along with their concentrations greatly contribute to the antimicrobial activity against Gram-positive and Gram-negative bacteria [55,56], this study attempted to define the antimicrobial activity of AgNPs engineered during green synthesis by taking advantage of commercially available synthetic *t*-FA and that of natural origin successfully isolated from RB after EH. The obtained results on the diameters of the inhibition zones are depicted in Table 5 and Figure 7. As seen, almost all substances exhibited remarkable suppression of bacteria growth. The most obvious inhibition for all tested test strains was achieved by AgNPs synthesized with biosynthetic *t*-FA and *t*-*iso*-FA isolates of RB.

### 3.5. Antimicrobial Activity of Silver Nanoparticles

The AgNPs engineered with biosynthetic FA displayed relatively better inhibition activity against Gram-negative pathogenic bacteria due to the smaller size of NPs and hence larger surface area that likely influenced the diffusion through the monolayer (2–3 mm) of peptidoglycan of the cell membranes more effectively than those of NPs with larger size [6,57]. However, the presence of bi- or multi-layers of peptidoglycan (30 mm) along with lipopolysaccharides in the cell walls of Gram-positive bacteria as reported by Miller et al. [58] makes them more rigid and resistant to environmental factors. Meanwhile, such MDR Gram-positive bacteria as *E. faecalis* display subpopulations with different surface charges, which manifested as bimodal zeta potential distribution [59].

This fact makes AgNPs adhesion and diffusion through cell membranes more challenging. However, the interactions of positively charged biocomposites along with bacteria damage mechanisms were well-documented [60,61], highlighting the ability of positively charged nano-scale objects to interact with negatively charged cell walls more effectively, contributing to electron transport chain interruption and the formation of ROS [62]. Meanwhile, the report of Rajendran et al. [63] indicated the ability of AgNPs and AuNPs mediated by FA derived from water extract of *Suaeda maritima* to promote ROS generation and cause DNA fragmentation of K562 cells without wherein inducing cell death of peripheral blood lymphocytes, thus providing a platform to combat leukemia. Encouraging results were reported by You et al. [64] indicating the ability of fabricated copper-tannic acid nanosheets grafted onto negatively charged sodium alginate used as hydrogel dressing to cause severe breakage of Gram-positive *S*. *aureus* cell membranes, leading to bacterial death and promoting the process of diabetic wound healing. Activity comparable to AgNPs inhibitory against *E. faecalis* was observed for RB isolate rich in *t*-FA and *t*-*iso*-FA, which is most plausibly due to the availability of OH groups affecting membrane ion exchange and reducing ATF synthesis by depleting cell membrane. It is worth noting, though, that the antimicrobial activity of AgNPs to some extent depends on their ability to induce the production of intracellular reactive oxygen species (ROS), which leads to oxidative stress in bacteria. This statement was reinforced by Mammari et al. [65], observing alteration of glycoproteins and lipids on the cytomembrane of MDR *P*. *aeruginosa* caused by copper indium sulfide/zinc sulfide (CuInS2/ZnS) quantum dots during exposure for 3 h. The effectiveness to induce membrane damage was found to be time- and concentration-dependent.

The most pronounced antimicrobial activity was observed for *E*. *coli* and *S*. *aureus*, where AgNPs were produced by biosynthetic *t*-FA and *t*-*iso*-FA, demonstrating superiority over AgNPs synthesized by the synthetic *t*-FA. The observed values are consistent with those of Tekin et al. [66] and Dalir et al. [55], highlighting the outstanding activity of AgNPs with an average size of 20 nm. However, with respect to the insights reported by Kourmouli et al. [67], the Kirby-Bauer method gives only a vague representation of the antimicrobial activity of AgNPs, which is in no way associated with their size, but rather with the ability of Ag to release ions under oxidation conditions. The antimicrobial activity assessment, based on direct contact of AgNPs by taking advantage of MIC and MBC approaches was conducted for comparative purposes (Table 6).

As expected, among the five test cultures evaluated, the most remarkable antimicrobial activity was observed for AgNPs engineered by biosynthetic *t*-FA and *t*-*iso*-FA, followed by AgNPs produced by synthetic *t*-FA. The MIC values varied in the range from 0.04 to 0.17 mg mL^−1^ and from 0.05 to 0.39 mg mL^−1^, respectively. The most remarkable contribution of AgNPs to the inhibition of bacteria was noticed for those mediated by biosynthetic *t*-FA and *t*-*iso*-FA demonstrating 0.04 mg mL^−1^ MIC values for *E*. *coli* and *P*. *aeruginosa*. Observed effective concentrations of AgNPs are consistent with those of [15,55], reinforcing the importance of the size of synthesized NPs. The highest concentration of AgNPs in the amount of 0.17 mg mL^−1^ has been required to inhibit the growth of *S*. *aureus*, though the amount found effective was 3.7-fold lower than reported by Parvekar et al. [68] for AgNPs of 5 nm. The effectiveness of engineered AgNPs in relation to *E. faecalis* inhibition should also be noted, as only 0.09 and 0.34 mg mL^−1^ were necessary for inhibition and complete death of this MDR pathogen, respectively.

Overall, the AgNPs engineered by green synthesis using biosynthetically produced *t*-FA and *t*-*iso*-FA, except for *P*. *aeruginosa*, were found to be more effective considering the values of MBC than those produced by synthetic *t*-FA. *P*. *aeruginosa* and *S*. *aureus* were found to be the most resistant to AgNPs as 0.69 mg mL^−1^ AgNPs were needed to cause irreversible consequences that resulted in death. The resistance of *P*. *aeruginosa* is due to the strong ability of this MDR species to produce a biofilm that protects them from harm as reported by Korzekwa et al. [7]. The observed data are consistent with those of El Din et al. [69], highlighting exceptionally low susceptibility of clinical MDR isolate *P. aeruginosa* to the AgNPs with observed MIC and MBC values as high as 8.0 mg mL^−1^.

## 4. Conclusions

Due to the limited availability of evidence on the fabrication of AgNPs using FA as a reducing agent, and given the need to reduce problems of environmental pollution with cereal by-products, this study demonstrated a technological solution of RB rational by taking advantage of enzyme-assisted hydrolysis. The abundance of both *t*-FA and *t*-*iso*-FA in the hydrolysate and eluate fraction obtained after EH and SPE, respectively, has been confirmed chromatographically. The results showed that the RB can be considered to be a cheap and renewable source of both *t*-FA and *t*-*iso*-FA, which as part of further green synthesis greatly contributed to the formation of AgNPs with proven antimicrobial activity against MDR pathogens. However, to enhance the antimicrobial activity of AgNPs, it is necessary to use NaOH, as its plays the role of an accelerator in the formation of AgNPs in the green synthesis. Meanwhile, the incorporation of amphiphilic PVP-40 as a capping and additional reducing agent greatly facilitated the formation of AgNPs with an average size of 16.5 nm. The fabricated AgNPs have shown effectiveness against both Gram-positive and Gram-negative pathogenic bacteria supported by the values of MIC and MBC. However, the ability of MDR *P*. *aeruginosa* to form a strong matrix of biofilm and the positive surface charge of *S*. *aureus* perhaps were the main factors that contributed to the relatively stronger resistance to the fabricated AgNPs.

To summarize, the introduced route of AgNPs synthesis is an environmentally friendly alternative with a safer profile than a chemical one, and could represent the future for sustainable industrial-scale NPs production. The elaborated AgNPs can be considered as active pharmaceutical ingredients (APIs) to be used for developing new antimicrobial agents and modifying therapies in treating MDR pathogens. However, the issue of AgNPs biosafety and potential health impacts in their various ways of exposure still remains relevant and needs to be covered.

## Figures and Tables

**Figure 1 nanomaterials-12-03053-f001:**
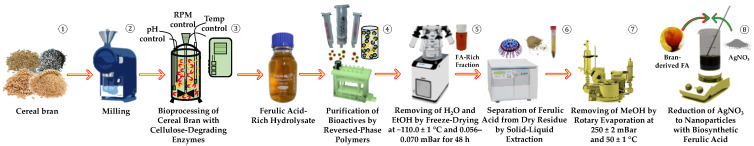
A schematic representation of major sample preparation steps undertaken for isolation of *trans*-FA and *trans*-*iso*-FA from rye bran for further green synthesis of silver nanoparticles.

**Figure 2 nanomaterials-12-03053-f002:**
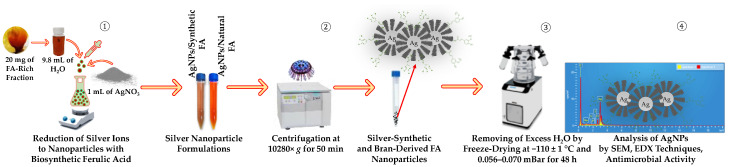
A schematic representation of major steps undertaken for the preparation of silver nanoparticles following the green synthesis principle using rye bran-derived *trans*-ferulic acid recovered by enzymatic hydrolysis with cellulose-degrading enzyme Viscozyme L.

**Figure 3 nanomaterials-12-03053-f003:**
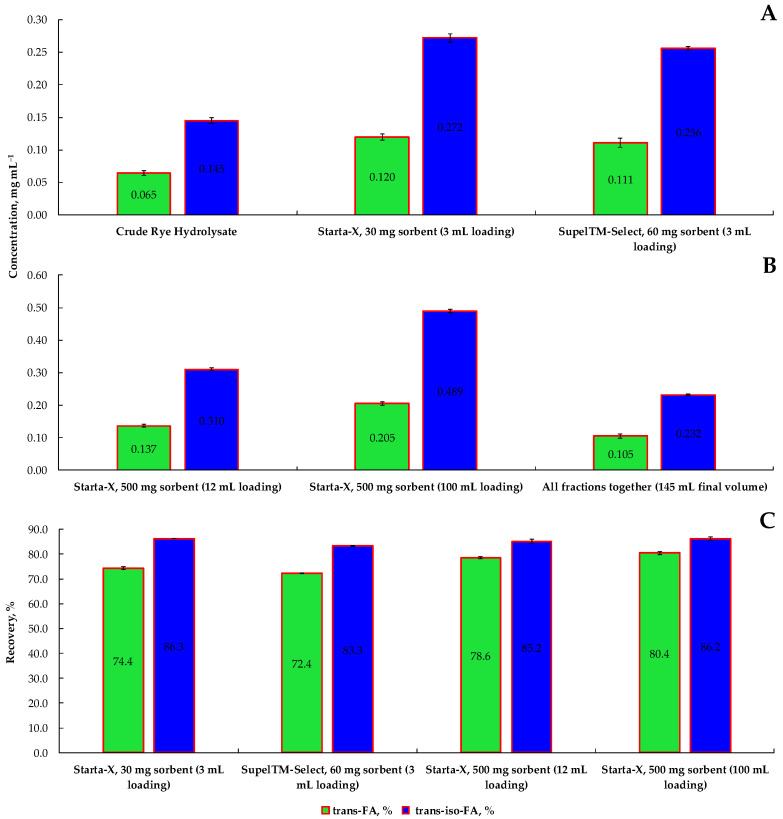
The concentration of *trans*-ferulic acid and *trans*-*iso*-ferulic acids obtained using enzymatic hydrolysis of rye bran with subsequent purification process conducted within small-scale (**A**) and large-scale (**B**) solid-phase extraction process using styrene-divinylbenzene-based reversed-phase sorbents of different masses and hydrolysate loading volumes, mg 100.0 mL^−1^ of eluate fraction. Relative recoveries of *trans*-ferulic acid and *trans*-*iso*-ferulic acids (%) from styrene-divinylbenzene-based reversed-phase sorbents (**C**).

**Figure 4 nanomaterials-12-03053-f004:**
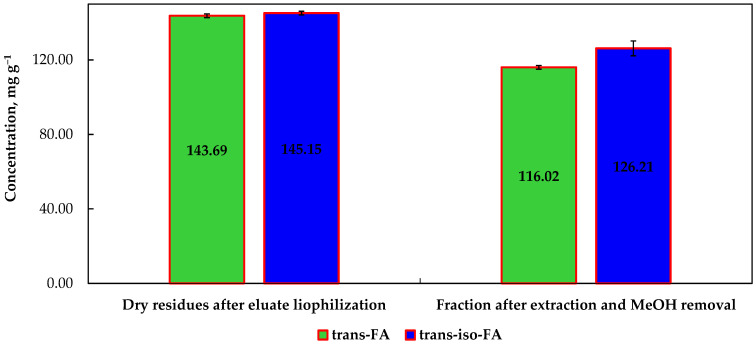
The amount of *trans*-ferulic acid and *trans*-*iso*-ferulic acids in the dry fraction obtained after EtOH evaporation and in the final fraction after MeOH removal, mg g^−1^.

**Figure 5 nanomaterials-12-03053-f005:**
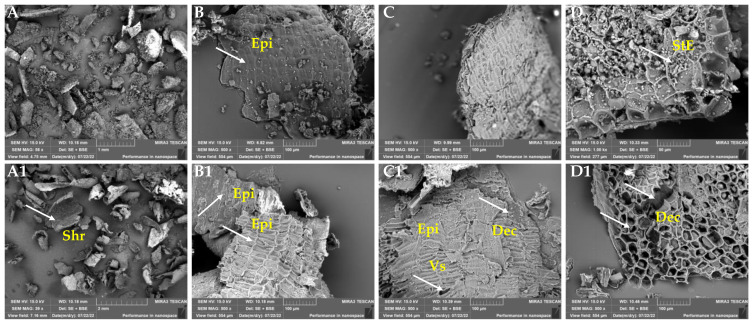
Scanning electron microscopy (SEM) images of untreated (**A**,**B**,**C**,**D**) and enzymatically hydrolyzed (**A1**,**B1**,**C1**,**D1**) rye bran samples. The hydrolysis was accomplished by cellulose-degrading commercial multi-enzyme complex Viscozyme L for 48 h. Note: white arrows indicate: smooth and homogeneous surface of bran epidermal layer (**B**); two fractions of starch granules embedded into starchy endosperm matrix (**D**); visible signs of bran shrinkage and twisting (**A1**); visible shrinkage of bran epidermal layer (**B1**); cracking (irregular void spaces) and partial decomposition of epidermal layer; tangible decomposition of the epidermal layer and ellipse-shaped void spaces of 40–50 µm in diameter with no starch granules (**D1**). Epi—epidermal layer; StE—starchy endosperm; Shr—shrinkage; Dec—decomposition; Vs—void spaces.

**Figure 6 nanomaterials-12-03053-f006:**
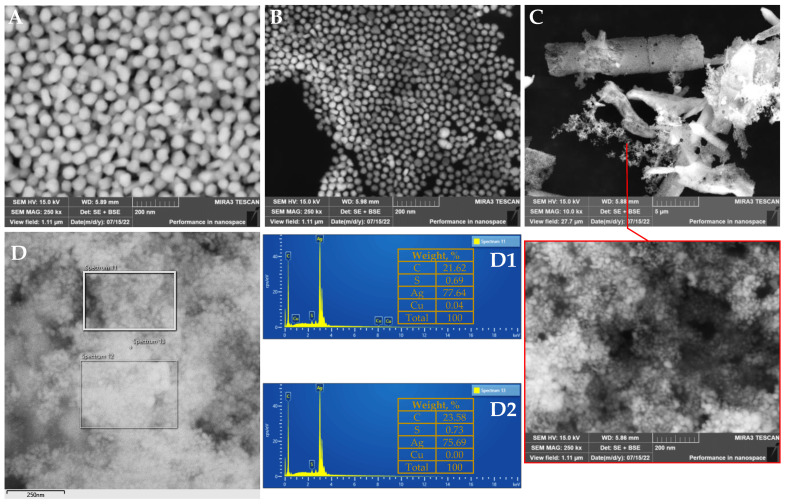
Scanning electron microscopy (SEM) images of uncapped (**A**) and capped (**B**) with polyvinylpyrrolidone synthesized silver nanoparticles produced by synthetic *trans*-ferulic acid, capped silver nanoparticles produced by rye bran-derived *trans*-ferulic acid and *trans*-*iso*-ferulic acids (**C**). EDS patterns (**D**,**D1**,**D2**) of silver nanoparticles produced by rye bran-derived *trans*-ferulic acid and *trans*-*iso*-ferulic acids captured at two randomly selected regions for EDS analysis.

**Figure 7 nanomaterials-12-03053-f007:**
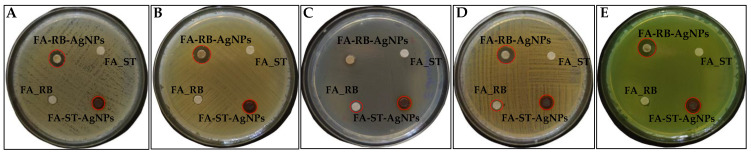
Disk diffusion tests for silver nanoparticles obtained using both synthetic and rye bran-derived *trans*-ferulic acid against *Bacillus subtilis* ATCC6633 (**A**), *Escherichia coli* ATCC25922 (**B**), *Enterococcus faecium* ATCC29212 (**C**), *Staphylococcus aureus* ATCC65388/NCTC7447 (**D**), and *Pseudomonas aeruginosa* ATCC10145 (**E**). The zone of inhibition is highlighted with a dashed red circle indicating a noticeable antibacterial effect. FA-ST-AgNPs—silver nanoparticles produced by synthetic *trans*-ferulic acid; FA-RB—a mixture of *trans*-ferulic acid and *trans*-*iso*-ferulic acids recovered from rye bran after enzymatic hydrolysis with multi-enzyme complex Viscozyme L; FA-RB-AgNPs—silver nanoparticles produced by biosynthetic *trans*-ferulic and *trans*-*iso*-ferulic recovered from rye bran after enzymatic hydrolysis with multi-enzyme complex Viscozyme L; FA-ST—synthetic *trans*-ferulic acid.

**Table 1 nanomaterials-12-03053-t001:** Nutritional compositions of cereal bran by-product derived from rye grains, g 100 g^−1^ DW.

	Major Nutrient Profile, g 100 g^−1^ DW					
Type of Bran	Moisture, %	Crude Carbohydrates	Starch	Crude Lipids	Crude Proteins	Dietary Fiber
Rye	11.7 ± 0.2 ^a^	30.9 ± 0.5 ^b^	18.6 ± 0.0 ^b^	3.8 ± 0.1 ^c^	16.9 ± 0.5 ^a^	36.0 ± 1.9 ^b^

Note: Values are means ± SD values of triplicates (*n* = 3). Means within the same column with different superscript letters (^a–c^) are significantly different at *p* ≤ 0.05; DW—dry weight; SD—standard deviation.

**Table 2 nanomaterials-12-03053-t002:** Main and side activities of multi-enzyme cellulose-degrading preparation used in this study.

Commercial Enzyme	Declared Activity	Enzyme Activity	Source	EC Number
Viscozyme^®^ L	100 FBG g^−1^	**Endo-1,3-(1,4)-β-d-glucanase**, Endo-1,4-β-xylanase, Non-reducing end α-L-arabinofuranosidase	*Aspergillus aculeatus*	3.2.1.6 3.2.1.8 3.2.1.55

Note: The main activities of enzyme used in this study are highlighted in bold. EC—enzyme commission; FBG—fungal β-glucanase units.

**Table 3 nanomaterials-12-03053-t003:** The MRM transitions and corresponding collision energy, Q1, Q3 and dwell time for investigated rye bran-derived bioactives.

Optimized MRM Parameters									Parameters of Calibration			
Organic Acid	RT, Min	Molecular Formula	Ionization Mode	MRM Transitions	Q1 Pre Bias, V	Collision Energy, V	Q3 Pre Bias, V	Dwell Time, Msec	R^2^	RSD, %	LOD, mg mL^−1^	LOQ, mg mL^−1^
*t*-FA	2.990	C_10_H_10_O_4_	[M−H]^−^	192.9500→134.0000	19.0	18.0	12.0	45.0	0.9996	5.81	0.004	0.014
				192.9500→178.0500	14.0	15.0	10.0	45.0				
*t*-*iso*-FA	2.962		[M+H]^+^	194.9000→177.1500	−12.0	−11.0	−20.0	45.0	0.9998	1.20	0.018	0.056
				194.9000→89.1500	−21.0	−33.0	−19.0	45.0				

Note: The first MRM transitions found were used for quantitative analysis, while the second for qualitative. *T*-FA—*trans*-ferulic acid; *t*-*iso*-FA—*trans*-*iso*-ferulic acid; RSD—relative standard deviation; LOD—limit of detection; LOQ—limit of quantification; RT—retention time; mg—milligram.

**Table 4 nanomaterials-12-03053-t004:** The concentration of mono- and disaccharides after enzymatic hydrolysis of rye bran for 48 h using multi-enzyme complex Viscozyme L, mg mL^−1^.

Carbohydrate	Control	RB Hydrolysate	Flow-Through Fraction after SPE
Gly	0.0 ± 0.0 ^b^	6.2 ± 0.2 ^a^	6.0 ± 0.1 ^a^
Xyl	0.3 ± 0.0 ^b^	4.9 ± 0.1 ^a^	4.6 ± 0.2 ^a^
Ara	0.4 ± 0.0 ^b^	1.6 ± 0.0 ^a^	1.4 ± 0.0 ^a^
Fru	n.d.	1.7 ± 0.0 ^a^	1.4 ± 0.0 ^a^
Glu	n.d.	10.7 ± 0.4 ^a^	9.0 ± 0.3 ^b^
Unk	n.d	0.4 ± 0.0 ^a^	0.4 ± 0.0 ^a^
Gala	n.d	1.2 ± 0.0 ^a^	1.3 ± 0.0 ^a^
Suc	1.8 ± 0.1 ^a^	0.8 ± 0.0 ^b^	0.9 ± 0.0 ^b^
Mal	0.2 ± 0.0 ^b^	1.6 ± 0.0 ^a^	1.4 ± 0.0 ^a^
Lac	n.d	0.9 ± 0.0 ^a^	0.6 ± 0.0 ^a^
Tot	2.7 ± 0.1 ^c^	30.0 ± 0.7 ^a^	27.0 ± 0.6 ^b^

Note: Values are means ± SD of triplicates (*n* = 3). Gly—glycerol; Xyl—xylose; Ara—arabinose; Fru—fructose; Glu—glucose; Unk—unknown; Gala—galactose; Suc—sucrose; Mal—maltose; Lac—lactose; Tot—total sugars content; RB—rye bran; SPE—solid-phase extraction; n.d.—not detected. The concentration of sugars in control RB sample is expressed as g 100 g^−1^ on a dry weight basis. Means within the same carbohydrate with different superscript letters (^a–c^) are significantly different at *p* < 0.05. Quantitative analysis of the unknown compounds was conducted according to the glucose calibration curve.

**Table 5 nanomaterials-12-03053-t005:** Antimicrobial activity of silver nanoparticles according to diameter of the inhibition zone values.

Reference Culture	Average Zone of Inhibition, mm			
	FA-ST-AgNPs	FA-RB	FA-RB- AgNPs	FA-ST
*Pseudomonas aeruginosa* ATCC 10145	10.0 ± 0.1 ^b^	5.2 ± 0.3 ^c^	11.3 ± 0.1 ^a^	5.0 ± 0.1 ^c^
*Enterococcus faecalis* ATCC 29212	8.0 ± 0.2 ^a^	7.1 ± 0.3 ^b^	7.5 ± 0.2 ^a^	5.1 ± 0.1 ^c^
*Escherichia coli* ATCC 25922	10.1 ± 0.4 ^b^	5.3 ± 0.6 ^c^	11.4 ± 0.1 ^a^	5.1 ± 0.1 ^c^
*Staphylococcus aureus* ATCC 6538P	8.2 ± 0.5 ^b^	6.2 ± 0.3 ^c^	11.2 ± 0.5 ^a^	5.3 ± 0.3 ^c^
*Bacillus subtilis* ATCC 6633	8.1 ± 0.1 ^b^	6.1 ± 0.2 ^c^	10.3 ± 0.4 ^a^	5.5 ± 0.4 ^c^

Note: Values are means ± SD values of duplicates (*n* = 2). Means within the same test culture with different superscript letters (^a–c^) are significantly different at *p* ≤ 0.05. FA-ST-AgNPs—silver nanoparticles produced by synthetic *trans*-ferulic acid; FA-RB—a mixture of *trans*-ferulic acid and *trans*-*iso*-ferulic acids recovered from rye bran after enzymatic hydrolysis with multi-enzyme complex Viscozyme L; FA-RB-AgNPs—silver nanoparticles produced by biosynthetic *trans*-ferulic and *trans*-*iso*-ferulic recovered from rye bran after enzymatic hydrolysis with multi-enzyme complex Viscozyme L; FA-ST—synthetic *trans*-ferulic acid.

**Table 6 nanomaterials-12-03053-t006:** Antimicrobial activity of AgNPs according to MIC and MBC (99.5%) values, mg mL^−1^.

Reference Culture	FA-ST-AgNPs		FA-RB		FA-RB-AgNPs		FA-ST	
	MIC	MBC	MIC	MBC	MIC	MBC	MIC	MBC
*Pseudomonas aeruginosa* ATCC 10145	0.05 ± 0.00 ^c^	0.20 ± 0.01 ^d^*	5.58 ± 0.17 ^a^	5.58 ± 0.43 ^a^*	0.04 ± 0.00 ^a^	0.69 ± 0.09 ^c^*	2.78 ± 0.17 ^b^	2.75 ± 0.24 ^b^*
*Enterococcus faecalis* ATCC 29212	0.39 ± 0.04 ^c^	3.13 ± 0.21 ^b^*	5.58 ± 0.32 ^a^	11.16 ± 1.76 ^a^*	0.09 ± 0.00 ^d^	0.34 ± 0.01 ^d^*	2.75 ± 0.23 ^b^	≥2.75 ^c^*
*Escherichia coli* ATCC 25922	0.10 ± 0.01 ^c^	0.39 ± 0.05 ^c^*	2.79 ± 0.28 ^a^	5.58 ± 0.45 ^a^*	0.04 ± 0.00 ^a^	0.17 ± 0.01 ^d^*	1.38 ± 0.15 ^b^	2.75 ± 0.19 ^b^*
*Staphylococcus aureus* ATCC 6538P	0.39 ± 0.02 ^c^	1.56 ± 0.07 ^b^*	5.58 ± 0.54 ^a^	11.16 ± 1.93 ^a^*	0.17 ± 0.00 ^c^	0.69 ± 0.09 ^c^*	1.38 ± 0.21 ^b^	1.38 ± 0.14 ^b^*
*Bacillus subtilis* ATCC 6633	0.20 ± 0.01 ^c^	0.78 ± 0.20 ^c^*	2.79 ± 0.15 ^a^	2.79 ± 0.13 ^a^*	0.04 ± 0.00 ^d^	0.09 ± 0.00 ^d^*	1.38 ± 0.18 ^b^	1.38 ± 0.21 ^b^*

Note: Values are means ± SD values of duplicates (*n* = 2). Means within the same test culture and method (MIC or MBC*) with different superscript letters (^a–d^) are significantly different at *p* ≤ 0.05. FA-ST-AgNPs—silver nanoparticles produced by synthetic *trans*-ferulic acid; FA-RB—a mixture of *trans*-ferulic acid and *trans*-*iso*-ferulic acids recovered from rye bran after enzymatic hydrolysis with multi-enzyme complex Viscozyme L; FA-RB-AgNPs—silver nanoparticles produced by biosynthetic *trans*-ferulic and *trans*-*iso*-ferulic recovered from rye bran after enzymatic hydrolysis with multi-enzyme complex Viscozyme L; FA-ST—synthetic *trans*-ferulic acid.

## Data Availability

The data sets and analysis of the study are available from the corresponding author upon reasonable request.

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
