# Peer review of "The Release of Non-Extractable Ferulic Acid from Cereal By-Products by Enzyme-Assisted Hydrolysis for Possible Utilization in Green Synthesis of Silver Nanoparticles"

_nanomaterials, 2022, doi:10.3390/nano12173053_

Round 1
Reviewer 1 Report
The current study reported a biosynthetic method for silver nanoparticles, and their antimicrobial activity was tested toward gram-positive and gram-negative bacteria. The experiment was designed properly and provided very useful data for future study. The current manuscript is fit for publication in Nanomaterials. I have only some suggestions below.
(1) The constructed AgNPs exhibited antimicrobial activity toward bacteria. I suggest authors to discuss the biosafety of AgNPs.
(2) Readers may want to know the difference between common AgNPs and current constructed AgNPs. For instance, the particle size, zeta potential or character?
(3) What is the advantage of current biosynthetic method? I recommend authors to provide the production cost.
(4) I suggest authors to discuss the potential application of AgNPs.
Author Response
Response to Reviewer’s 1 comments
R: The current study reported a biosynthetic method for silver nanoparticles, and their antimicrobial activity was tested toward gram-positive and gram-negative bacteria. The experiment was designed properly and provided very useful data for future study. The current manuscript is fit for publication in Nanomaterials. I have only some suggestions below.
A: The authors would like to thank the reviewer for carefully checking our manuscript and for your valuable comments. In preparing the manuscript authors have incorporated most of the changes suggested. The authors refer to them in detail below.
R: The constructed AgNPs exhibited antimicrobial activity toward bacteria. I suggest authors to discuss the biosafety of AgNPs
A: Dear reviewer! We find it difficult to operate with information that has not been supported by the current study. Based on the information currently available: Rajendran, et al. doi:10.1080/21691401.2016.1236803, and considering the way the AgNPs were produced using pure plant-derived compounds rather than crude extracts, the authors can only hypothesize that the synthesized nanoparticles can be considered safe. The biosafety of FA-AgNPs will be elucidated in our further studies. Additional discussion on this matter was provided within lines 588-592.
R: Readers may want to know the difference between common AgNPs and current constructed AgNPs. For instance, the particle size, zeta potential or character?
A: Dear reviewer! The information regarding particle size has been provided in lines 25, 28, 30, 538, Meanwhile, the authors already made a comparison of the particle size of AgNPs mediated by biosynthetically obtained FA with those engineered by synthetic FA (lines 521), extract of Eucalyptus camaldulensis leaves (line 547), trisodium citrate (lines 549-550), eugenol and Carya illinoinensis leaf extract (lines 613-614) as reducing agents.
R: What is the advantage of current biosynthetic method?....
A: The advantages of the designed enzyme-assisted AgNPs over other nano-scale objects are conditioned by the fact that in the process of green synthesis the way problematic by-products, i.e., CB was used as a source of natural reducing agents. Given the need to reduce problems of environmental pollution with grain processing by-products, in particular, the present study demonstrated a technological solution of by-products rational use as a potential source of bioactives with functional groups in the sustainable production of metallic NPs. The technology does not envisage the use of either toxic or expensive chemicals while generating environmentally benign highly demanded products. The synthesis of nano-scale objects through the bioreduction route currently provides a platform to combat pathogenic bacteria in the era of antimicrobial resistance. Their adverse effect on pathogenic microorganisms has been well documented. However, the nano-toxic potential of metallic NPs engineered with crude extracts in vivo studies within human clinical trials still remains unclear. It has been proposed that the crude extracts in AgNPs synthesis could lead to the formation of undesirable complexes in the final products resulting in limiting the range of AgNPs applicability. This is due to the complexity of crude extracts that are replete with components of various natures. Taking this into account, the current study provided insight into the possible utilization of individual components of plant-derived origin, i.e., t-FA and t-iso-FA in the green synthesis of AgNPs without taking a risk in the formation of side products except for quinones.
R: …I recommend authors to provide the production cost.
A: The authors understand the reviewer's concern regarding AgNPs production costs. However, at the moment, the authors cannot provide such information, since neither optimization nor a scale-up process of AgNPs synthesis was carried out. Since the operational and process conditions can have a direct impact on the response variables such as the yield and morphological features of synthesized AgNPs resulting in NPs properties change. Therefore, as part of our next steps, an optimization process will be carried out supporting the Face Centered Central Composite Design (FCCCD) while considering four parameters, i.e., AgNO3 and NaOH concentrations, pH, and temperature of the reaction.
R: I suggest authors to discuss the potential application of AgNPs.
A: The authors appreciate the reviewer’s valuable suggestion. Additional information indicating the potential utilization of synthesized AgNPs was provided in the Abstract (lines 37-39) and Conclusions (lines 658-660) sections.
On behalf of all the co-authors
Yours sincerely,
Vitalijs Radenkovs
Principal investigator, Latvia University of Life Sciences and Technologies, Research Laboratory of Biotechnology, Division of Smart Technologies.
Reviewer 2 Report
1. What are the advantages of the designed enzyme-assisted silver nanoparticles in this paper compared to other systems? The comparison and the advantages can be added to the manuscript.
2. Figure 7, is there an improvement in the antibacterial performance of the enzyme strategy preparation compared to other nanosystems?
3. The antibacterial mechanism could be improved by citing: 10.1016/j.mtadv.2022.100271;10.1016/j.mtbio.2022.100264;10.1016/j.cej.2022.135691. 10.1021/acsami.1c25014
4 How does the cost of enzyme strategy preparation compare to conventional preparation? Is there a potential for large-scale application?
5. It could be better if a brief comment (challenges and future prospects) is added at the end of the conclusion.
Author Response
Response to Reviewer’s 2 comments
A: The authors would like to thank the reviewer for carefully checking our manuscript and for your valuable comments. In preparing the manuscript authors have incorporated most of the changes suggested. The authors refer to them in detail below.
R: 1. What are the advantages of the designed enzyme-assisted silver nanoparticles in this paper compared to other systems?....
A: Dear reviewer! The advantages of the designed enzyme-assisted AgNPs over other nano-scale objects are conditioned by the fact that in the process of green synthesis the way problematic by-products, i.e., CB was used as a cheap and renewable source of natural reducing agents. Given the need to reduce problems of environmental pollution with grain processing by-products, in particular, the present study demonstrated a technological solution of by-products' rational use as a potential source of bioactives with functional groups in the sustainable production of metallic NPs. The technology does not envisage the use of either toxic or expensive chemicals while generating environmentally benign demanded products. The synthesis of nano-scale objects through the bioreduction route currently provides a platform to combat pathogenic bacteria in the era of antimicrobial resistance. Their adverse effect on pathogenic microorganisms has been well documented. However, the nano-toxic potential of metallic NPs engineered with crude extracts in vivo studies within human clinical trials still remains unclear. It has been proposed that the crude extracts in AgNPs synthesis could lead to the formation of undesirable complexes in the final products resulting in limiting the range of AgNPs applicability. This is due to the complexity of crude extracts that are replete with components of various natures. Taking this into account, the current study provided insight into the possible utilization of individual components of plant-derived origin, i.e., t-FA and t-iso-FA in the green synthesis of AgNPs without taking a risk in the formation of side products except for quinones.
R: ….The comparison and the advantages can be added to the manuscript.
A: The authors appreciate the reviewer's suggestion to include an additional comparison of engineered AgNPs to already existing nano-objects. Additional comparison has been provided within lines 595-610.
R: 2. Figure 7, is there an improvement in the antibacterial performance of the enzyme strategy preparation compared to other nanosystems?
A: Comparing the data with already published literature, CB-derived t-FA and t-iso-FA used in this study as reducing agents in the green synthesis of AgNPs delivered comparable or even better results regarding the antimicrobial activity against certain species of pathogens. The obtained zone of inhibition values are consistent with those reported for Au NPs (10.2147/IJN.S119618; 10.1016/j.jsps.2018.11.010), ZnO NPs (10.1007/s10867-019-9520-4; 10.3390/molecules27113532), spider silk proteins and mesoporous silica nanoparticles loaded with antibiotics and antimycotics (10.1002/marc.201900426), however, substantially lower than reported for CuNPs (10.1016/j.aoas.2021.01.006). Meanwhile, it has been observed that AgNPs mediated by biosynthetically obtained t-FA and t-iso-FA were substantially smaller in size with homogenous shape than that of AgNPs mediated by synthetic t-FA. The presence of additional hydroxyl moieties ensured by t-iso-FA favorably affected the positive charging of AgNPs and promoted their more uniform distribution over the surface of bacteria causing irreversible consequences. However, more detailed studies are needed to confirm these statements.
R: 3. The antibacterial mechanism could be improved by citing: 10.1016/j.mtadv.2022.100271;10.1016/j.mtbio.2022.100264;10.1016/j.cej.2022.135691. 10.1021/acsami.1c25014
A: The authors appreciate the reviewer's suggestion to include additional references to the manuscript very much. The authors have supplemented the manuscript with additional information supported by 10.1016/j.mtadv.2022.100271.
R: How does the cost of enzyme strategy preparation compare to conventional preparation? Is there a potential for large-scale application?
A: From a technical point of view, the technology for the production of ferulic acid has a place to be and can be adapted by the industry. However, one should consider the initial investments that will be required for its introduction. Most of the costs can be related to the acquisition of a semi-batch reactor needed for the hydrolysis of cellulose and hemicellulose fractions, as well as sorbent columns for the purification/ concentration of hydrolysate. To save costs and reduce material losses flow process should be ensured. Meanwhile for the synthesis of NPs the same batch reactor could be used following “wet chemical” process, i.e., synthesis → precipitation → separation → washing → sedimentation → evaporation. Considering the green synthesis of NPs using plant extracts, the process can proceed differently and out of control, each time giving different response variables such as the yield, size, and shape of NPs which consequently will affect their properties. In turn, the considered technology involves the use of RB fraction containing mostly t-FA and t-iso-FA. The current market value of t-FA and t-iso-FA is approximately €169.00 and €1800 per 100 g, respectively. The high price is because it is produced mostly by chemical synthesis at industrial-scale. The manufacturing of FA in this way is reported to be a laborious and time-consuming process (up to three weeks) that requires a significant input of conditionally toxic reagents (malonic acid), solvents (piperidine), and catalysts (benzylamines). The production of AgNPs on a large-scale using FA as a reducing agent becomes quite realistic if it includes cereal bran-derived FA. However, to validate this, optimization and scale-up process of AgNPs synthesis are to be performed.
R: It could be better if a brief comment (challenges and future prospects) is added at the end of the conclusion.
A: A brief comment on challenges and further prospects has been added as proposed by the reviewer.
On behalf of all the co-authors
Yours sincerely,
Vitalijs Radenkovs
Principal investigator, Latvia University of Life Sciences and Technologies, Research Laboratory of Biotechnology, Division of Smart Technologies.
Reviewer 3 Report
Green synthesis of silver nanoparticles with reagents originated from natural resources is important for the sustainable development. The research of this manuscript is interesting and results are reliable. However, revisions are required and the comments are given below.
1. Green synthesis of silver nanoparticles is an interesting topic and many good papers have been published. Some close related references are suggested to be cited, for example “Preparation and properties of cellulose nanocomposite fabrics with in situ generated silver nanoparticles by bioreduction method; Nanocomposite egg shell powder with in situ generated silver nanoparticles using inherent collagen as reducing agent”.
2. Please pay attention to the writing of subscripts and superscripts, for example “AgNO3” in line 234, “mg mL-1” in line 340, “OH- and COOH-” in line 500 need to be fixed.
3. What does “Beefily” in line 319 means? Are the authors want to say “Briefly”? “coper” in line 519 need to be fixed. The authors are suggested to go through the manuscript again to reduce typos.
4. The numbers in Figure 6 are hard to tell. Please remove them.
5. Please check the references, some references lack page numbers. Also pay attention to the subscripts of some chemical reagents.
Author Response
Response to Reviewer’s 3 comments
R: Green synthesis of silver nanoparticles with reagents originated from natural resources is important for the sustainable development. The research of this manuscript is interesting and results are reliable. However, revisions are required and the comments are given below.
A: The authors would like to thank the reviewer for carefully checking our manuscript and for your valuable comments. In preparing the manuscript authors have incorporated most of the changes suggested. The authors refer to them in detail below.
R: Green synthesis of silver nanoparticles is an interesting topic and many good papers have been published. Some close related references are suggested to be cited, for example “Preparation and properties of cellulose nanocomposite fabrics with in situ generated silver nanoparticles by bioreduction method; Nanocomposite egg shell powder with in situ generated silver nanoparticles using inherent collagen as reducing agent”.
A: The authors have included the provided reference materials in their manuscript as proposed.
R: Please pay attention to the writing of subscripts and superscripts, for example “AgNO3” in line 234, “mg mL-1” in line 340, “OH- and COOH-” in line 500 need to be fixed.
A: The authors proofread their manuscript and such typos were fixed.
R: What does “Beefily” in line 319 means? Are the authors want to say “Briefly”? “coper” in line 519 need to be fixed. The authors are suggested to go through the manuscript again to reduce typos.
A: The authors are sorry for misleading the reviewer with such typos. The authors have made the necessary corrections
R: The numbers in Figure 6 are hard to tell. Please remove them.
A: The authors are grateful to the reviewer for his valuable remark. The mentioned figure was amended.
R: Please check the references, some references lack page numbers. Also pay attention to the subscripts of some chemical reagents.
A: Dear reviewer! Some online-only Publishers (e.g. Biomed Central) recently started to use article numbers instead of page numbers. For such articles without page numbers, reference to the article number is provided.
On behalf of all the co-authors
Yours sincerely,
Vitalijs Radenkovs
Principal investigator, Latvia University of Life Sciences and Technologies, Research Laboratory of Biotechnology, Division of Smart Technologies.
Round 2
Reviewer 2 Report
I don't think it needs any further modification